# The Relationship between Participation in Extracurricular Arts and Sports Activities and Adolescents’ Social and Emotional Skills: An Empirical Analysis Based on the OECD Social and Emotional Skills Survey

**DOI:** 10.3390/bs14070541

**Published:** 2024-06-27

**Authors:** Weihao Wang, Wenye Li, Jijun Yao

**Affiliations:** 1The Faculty of Education, East China Normal University, Shanghai 200062, China; wangweihao_ecnu@163.com; 2The Institute of Education, Nanjing University, Nanjing 210023, China; nju_yeahlee@foxmail.com; 3School of Education Science, Nanjing Normal University, Nanjing 210024, China

**Keywords:** extracurricular activity, social and emotional skills, coarsened exact matching (CEM)

## Abstract

In light of the ‘double reduction’ policy, which affords adolescents increased time for extracurricular pursuits, the strategic organization of these activities’ form and content is imperative. Prior research has established a robust correlation between adolescent participation in extracurricular arts and sports and the enhancement in their social and emotional skills. Nevertheless, the relationship between extracurricular arts and sports activities and the various dimensions of social and emotional skills, as well as the connection between participation in different types of these activities and the enhancement in social and emotional skills, requires further investigation. Utilizing the Theory of Multiple Intelligence and data from the OECD-SSES2019 Suzhou (China) student survey, this study employs Ordinary Least Squares (OLS) and coarsened exact matching (CEM) methodologies to address these gaps. In China, participation in extracurricular arts and sports activities is significantly positively associated with various dimensions of social and emotional skills, with a synergistic effect observed between these activities in enhancing these skills. Additionally, this study finds age-related heterogeneity in the relationship between participation in extracurricular arts and sports activities and the improvement in social and emotional skills.

## 1. Introduction

In the contemporary era, numerous countries are reassessing the purpose and value of education, prompted by novel paradigms in thinking, working, and living, as well as the necessity to adapt to a life paced faster and an environment more complex, all emerging from the information revolution [1,2,3]. Against this backdrop, the cultivation and development of adolescents’ social and emotional skills have ascended to a position of priority in the formulation of educational policies worldwide [4,5,6]. ‘Social and emotional skills’ denote an individual’s capacity to self-regulate thoughts, emotions, and behaviors, serving as a critical predictor of academic, career, health, and overall well-being outcomes [1,7]. Distinct from cognitive competencies, which focus on enhancing an individual’s information processing and problem-solving abilities, social and emotional skills emphasize the management of emotions and the effective perception of one’s existence and interactions with others [1,8,9].

In China, education scholars and policymakers have been concerned about the issue that exam-oriented education places too much emphasis on cognitive training for students while overlooking their all-round development. In July 2021, the General Office of the Central Committee of the Communist Party of China (CPC) and the General Office of the State Council issued “Opinions on Further Reducing the Burden of Students’ Homework and Off-campus Training in Compulsory Education”, proposing to “improve the quality of after-school services and meet the diverse needs of students”, requiring that “schools should make full use of resource advantages, effectively implement various after-school education activities, and meet students’ diverse learning needs in the school”, and at the same time “carry out various activities in popular science, sports, art, labor, reading, interest groups, and clubs” [10]. Admittedly, scientifically designing after-school service content to develop students’ social and emotional skills has become an important task for school education in China.

## 2. Literature Review

In recent years, social and emotional skills have caught the attention of researchers from all over the world [7,11,12]. Studies revealed that the cultivation of adolescents’ social and emotional skills was positively associated with their future academic performance, well-being, career achievement, and quality of life, and such skills played a vital role in coping with emergencies, calming anger, and seeking cooperation [13,14,15]. Other studies showed that developing adolescents’ social and emotional skills could improve their sense of responsibility and prosocial behaviors [16,17], and adolescents with better social and emotional skills were less likely to be involved in smoking, alcoholism, and bullying [18]. Therefore, social and emotional skills play a significant role in adolescents’ growth.

Some studies suggested that extracurricular activities provided a positive environment for adolescents’ growth [19,20,21], and participation in extracurricular activities promoted adolescents’ social and emotional skills [22,23]. Neely and Vaquera found that participation in extracurricular sports and arts activities significantly reduced dropout rates in high school students [24]. Pham et al. revealed that participation in extracurricular sports was correlated to life satisfaction based on the survey of 416 students aged 13 to 17 in 12 schools in Vietnam [25]. Feraco et al. conducted a study with a sample of 460 students aged between 5 and 12 and found that participating in extracurricular activities had an impact on students’ soft skills such as resilience, perseverance, curiosity, and leadership [26]. Apparently, participation in extracurricular arts and sports activities has a significant positive relationship with adolescents’ social and emotional skills.

At the same time, some other studies suggested that participating in extracurricular activities had an indecisive impact on the development of adolescents’ social and emotional skills. Baker noted that the impact of extracurricular activities on students is multifaceted, with the type of activity and gender of participants being key determinants [27]. The research project implemented by Ren and Zhang on a study of 194 preschool children in Hong Kong concluded that extracurricular activities could promote the development of children’s non-cognitive skills but were not related to the development of their social skills [28]. Therefore, different extracurricular activities may have varying relationships with the enhancement in social and emotional skills in adolescents from different backgrounds.

Many studies have also discovered that simultaneous participation in various types of extracurricular activities is more significantly linked to adolescent development, with higher related effects, suggesting a synergistic effect. The term “synergistic effect” refers to the phenomenon where a student’s participation in both extracurricular art and sports activities yields greater expected benefits than the sum of the expected benefits of participating in either extracurricular sports activities only or extracurricular arts activities only. Feldman and Matjasko divided adolescents in terms of extracurricular activities that they participated in into six mutually exclusive groups, namely participation in sports activities only, participation in academic activities only, participation in school activities only, participation in performance activities only, participation in multiple extracurricular activities, and participation in no activities [29]. The study results showed that the academic achievements of students who participated in multiple extracurricular activities were significantly higher than those of students in other groups. Blomfield and Barber divided the 1489 adolescents who participated in their study into the group with students only participating in sports extracurricular activities (such as basketball, swimming, etc.), the group with students only participating in non-sports extracurricular activities (such as bands, drama clubs, etc.), the group with students participating in both sports and non-sports extracurricular activities, and the group with students participating in no extracurricular activities [30]. Their findings indicated that students who participated in both sports and non-sports extracurricular activities performed significantly better than the other three groups in terms of self-social concept and self-worth perception, and they presented significantly better self-academic performance than students who only participated in sports extracurricular activities and those who did not participate in any extracurricular activities.

The social and emotional skills of adolescents may be affected not only by the type of extracurricular activities they participate in but also by gender [31,32,33,34,35], family socioeconomic background [36], parent–child relationship [37], peer relationship [38], and other factors. Regarding the gender factor, Wang and Mao’s research on the social and emotional skills of left-behind children (who are left in rural homes while their parents go to work in cities) found that boarding school had a more significant negative impact on boys [39]. Their study implied that gender as a variable should be included as a covariable in the studies of social and emotional skills [40,41]. Additionally, in terms of family socioeconomic background, extracurricular activities were contextual and rooted in the school and community environment and were influenced by and interacted with family- and peer-related factors [42].

The family’s socioeconomic background had a significant impact on students’ academic achievements, and their social and emotional skills [43,44,45]. Campos, Campos, and Barrett pointed out that parenting style was crucial for the development of children and adolescents [46]. Anthony et al.’s study suggested that negative parenting behaviors caused by parents’ pressure hurt their children’s development [47]. Zhu surveyed 965 Hong Kong adolescents and pointed out that parents played an irreplaceable role in the development of adolescents’ social and emotional skills [18]. His study also showed that there were gender differences in the relationship between parenting behavior and children’s social and emotional skills in that girls were more likely to be influenced by parenting behaviors.

Peer relationships play a crucial role in the development of adolescents [48]. Adolescents who feel cared for and valued by their peers are more likely to independently explore their learning and social environments, thereby enhancing their self-efficacy, academic performance, and social and emotional skills [49,50]. Conversely, experiences of peer rejection and aggression are associated with increased negative social emotions and decreased academic performance [51,52].

Based on the Theory of Multiple Intelligences (TMI), this paper presents research on the impact of participating in extracurricular sports and arts activities on the social and emotional skills of adolescents. TMI holds that the abilities required for individuals to solve real problems or to produce and create effective products are fundamentally diverse; i.e., there is not one such ability but a set of abilities. The basic structure of such abilities is also diverse with various abilities existing not in an integrated form but a relatively independent form [53,54]. The assessment of an individual’s intelligence should shift from traditional single intelligence testing to the assessments of a range of abilities. An individual has at least eight types of equally important intelligence, including language, mathematical logic, musical rhythm, visual space, physical movement, interpersonal communication, self-reflection, and nature observation, and such intelligence is manifested in different forms [55,56,57]. According to TMI, the abilities of each intelligence dimension are independent, but also connected in the process of their development. Informed by TMI, we argue that extracurricular sports and arts activities can help promote the development of adolescents’ social and emotional skills while developing their musical and rhythmic intelligence and bodily–kinesthetic intelligence.

Existing studies in China and other countries have examined the influence of various factors on adolescents’ social and emotional skills. However, several limitations persist. Firstly, many studies rely on case research or small samples, lacking robust conclusions from large-scale surveys. Secondly, the complex nature of social and emotional skills necessitates a detailed analysis of how different extracurricular activities impact personal development. Previous studies often oversimplify by treating these activities merely as broad variables under “social and emotional skills”, which is inadequate. Thirdly, there is room for improvement in the precision of these studies and the inclusion of control variables. To address these issues, this study utilizes data from the OECD-SSES2019 Suzhou (China) survey, ensuring scientific rigor and authority [1]. Additionally, this study employs the coarsened exact matching (CEM) method to mitigate sample selection bias and enhance analytical precision.

Notably, China is the country that experienced the largest decline in social and emotional skills in the groups of 10-year-olds and 15-year-olds among the nine countries that participated in the OECD assessment. To avoid the continued deteriorating situation in China, this study is designed to find out the effective means that can promote social and emotional skills. In accordance with the current research landscape and the aforementioned research inquiries, this study posits the following hypotheses:

**H1:** 
*Participation in extracurricular sports is significantly associated with the enhancement in adolescents’ social and emotional skills.*


**H2:** 
*Participation in extracurricular arts is significantly associated with the enhancement in adolescents’ social and emotional skills.*


**H3:** 
*There exists a synergistic effect between extracurricular sports and arts activities on the enhancement in adolescents’ social and emotional skills.*


**H4:** 
*The relationship between extracurricular sports and arts activities and the enhancement in adolescents’ social and emotional skills exhibits heterogeneity.*


## 3. Research Data and Methods

### 3.1. Data Source

We collected this study data from the Suzhou (China) part of the international large-scale survey that assesses adolescents’ social and emotional skills conducted by OECD in ten cities in nine countries around the world in 2019. Considering that Suzhou is an emblematic city in southeastern China, it is probable that other areas will experience similar scenarios and challenges as the Chinese economy progresses, and reach a comparable level of development to Suzhou. As a result, it is crucial to carry out a comprehensive evaluation and survey of the Suzhou pilot outcomes, as they can furnish valuable knowledge and understanding for these regions [58]. The OECD’s social and emotional skills study drew on the “Big Five” model to construct an assessment framework for social and emotional skills. This framework was mainly divided into five dimensions: task performance, emotional regulation, collaboration, open-mindedness, and engaging with others [58]. The OECD sample in Suzhou was divided into the 10-year-old group and the 15-year-old group. The exact age range for the 10-year-old group included students between 10 years and 2 months old and 11 years and 1 month old, whereas students for the 15-year-old group were between 15 years and 2 months to 16 years and 1 month. A total of 151 primary and secondary schools in 6 districts and 4 county-level cities under the jurisdiction of the city of Suzhou were formally evaluated. A total of 7268 students participated in the assessment, including 3647 students in the 10-year-old group and 3621 students in the 15-year-old group [58]. In this study, 7213 valid student samples (52.86% boys and 47.14% girls) were finally obtained through data sorting and matching and excluding invalid data.

### 3.2. Variables

To evaluate the efficacy of the scales used to measure students’ social and emotional skills, a survey employing assessment scale triangulation was conducted. The survey’s technical reports indicate that the social and emotional skills measures used in the Chinese cultural context demonstrated satisfactory psychometric properties, as evidenced by a Cronbach’s alpha value exceeding 0.7, standardized factor loading coefficients surpassing 0.4, all dimensions exhibiting CFIs greater than 0.90 (with the exception of the energy dimension among 10-year-old students, which exhibited a CFI of 0.890), and all SRMRs less than 0.06 [3,59]. According to OECD [1], collaboration was defined as “caring about the well-being of others” and included three sub-dimensions of empathy, trust, and cooperation [1,33,60]. Emotional regulation was defined as whether a person could manage emotions well and whether they could effectively adjust to negative emotions and deal with stress, and whether they had an optimistic attitude towards personal life and social career development. Emotional regulation included three sub-dimensions of stress resistance, optimism, and emotional control. Engaging with others referred to the ability of an individual to interact with others and establish and maintain social relationships [1,30,60]. Engaging with others was reflected as “extroversion” in the Big Five model and included three sub-dimensions of sociability, assertiveness, and energy [1,35,60]. Open-mindedness reflected the openness of the research participant’s mind. It depicted the individual’s willingness to try new things and accept new experiences. Reflected as “openness” in the Big Five model, open-mindedness included three sub-dimensions of tolerance, curiosity, and creativity [1,34,60]. Task performance was based on the ability of conscientiousness in the “Big Five” model, including three sub-dimensions of self-control, responsibility, and perseverance [1,32,60].

In this study, the three sub-dimensions of cooperation (COO), empathy (EMP), and trust (TRU) derived from the student questionnaires were added up and then averaged to obtain the collaboration score. The scores of the three sub-dimensions of emotional control (EMO), optimism (OPT), and stress resistance (STR) were added up and then averaged to obtain the score for the emotional regulation. The scores of the three sub-dimensions of sociability (SOC), assertiveness (ASS), and energy (ENE) were added up and then averaged to achieve the score for engaging with others. The scores of the three sub-dimensions of tolerance (TOL), curiosity (CUR), and creativity (CRE) were added up and then averaged to obtain the score for open-mindedness. The scores of three sub-dimensions of self-control (SEL), responsibility (RES), and perseverance (PER) were added up and averaged to attain the score for task performance. The technical documents and literature published by the OECD [3] and Huang [35] provide detailed information on the specific topics pertaining to each dimension.

Due to previous research conducted by the OECD revealing significant differences in the development of social and emotional skills between 10-year-old and 15-year-old students, it is necessary to separately analyze the factors and mechanisms that influence the development of social and emotional skills in these two age groups [3]. The grouping variable denotes whether students participate in extracurricular sports and arts activities. The survey question “Do you participate in any of the following extracurricular activities outside of school?” had primary items “Sports (e.g., clubs, lessons, etc.)” and “Arts (e.g., playing a musical instrument, dancing, drawing, etc.)” with response options “No” or “Yes”. A “No” indicates no participation in extracurricular sports or arts activities, while a “Yes” indicates participation. Students were categorized into four groups based on their responses: only participating in arts activities, only participating in sports activities, participating in both, and participating in neither. The analysis was divided into three groups. In the first set of analyses, a value of 1 was assigned to those who participated solely in extracurricular art activities (without participating in extracurricular sports activities), while a value of 0 was assigned to those who did not participate in either extracurricular art or sports activities. In the second set of analyses, a value of 1 was assigned to those who only participated in extracurricular sports activities (without participating in extracurricular art activities), while a value of 0 was assigned to those who did not participate in either extracurricular art or sports activities. In the third set of analyses, a value of 1 was assigned to those who participated in both extracurricular art and sports activities, while a value of 0 was assigned to those who did not participate in either extracurricular art or sports activities.

Control variables for this study included student gender, family socioeconomic background, peer relationships, and parent–child relationships. The gender of students was a dichotomous variable, with a value of 1 for boys and a value of 0 for girls. The family socioeconomic score (SES) is a continuous variable derived from parents’ educational levels, parents’ occupations, and family wealth data through a principal component analysis to obtain factor scores. These scores are then used to calculate the family socioeconomic background score, where a higher score indicates a better family socioeconomic background. The peer relationship is a continuous variable derived from four items: “My friends understand me”, “My friends accept me as I am”, “My friends are easy to talk to”, and “My friends respect my feelings”. The mean score of the four items is calculated, and a higher mean score indicates a better relationship between the student and their peers. The parent–child relationship is a continuous variable derived from three items: “I get upset easily with my parents”, “It is hard for me to talk with my parents”, and “I feel angry with my parents”. The mean score of the three items is calculated, and a higher mean score indicates a more disharmonious relationship between the parent and child.

The descriptive statistics of this study are shown in Table 1 below.

### 3.3. Methods

To avoid the biased results and estimates of multiple linear regression stemming from biased sample selection in social science research, this project adopted the CEM method based on OLS regression. The basic idea of the CEM method was to process each variable by re-encoding them so that values that could not have been grouped can be grouped and assigned weights. The CEM method adopted the “exact matching” algorithm to match the processed data, and the unmatched samples were excluded from matching samples. In this way, the processed data were restored, and the original data were kept [61].

Drawing from the practice of Li and Yao [62], this study yielded weights (CEM_weights) via the coarsened exact matching technique and brought them into the regression model for weighted regression to control the sample selection bias and reduce the imbalanced characteristic variables between groups. In this study, students who only participated in extracurricular arts activities, students who only participated in extracurricular sports activities, and students who participated in both extracurricular arts activities and sports activities were set as different treatment groups, whereas students who participated in neither extracurricular arts activities nor sports activities were set as the control group. Estimation bias caused by sample selection was controlled by matching samples in the treatment groups with samples from the control group that have similar characteristics in gender, family socioeconomic background, parent–child relationship, and peer relationship to investigate the differences in social and emotional skills among students who participated in different types of extracurricular sports and arts activities.

## 4. Empirical Results Analysis

This study adopted the multiple linear regression method to evaluate the impact of participating in extracurricular arts and sports activities on the various measurement dimensions of students’ social and emotional skills. This study found no collinearity and serious heteroscedasticity problems. As shown in Table 2, after controlling for relevant variables, participation in extracurricular arts and sports activities is significantly positively associated with improvements in various dimensions of social and emotional skills. The relationship between extracurricular activities and different dimensions of social and emotional skills varies across different measurement dimensions. The correlation coefficient for extracurricular sports activities is higher than that for extracurricular arts activities. Additionally, students who participate in both extracurricular arts and sports activities have higher correlation coefficients than those who engage in only one type of activity.

Table 3 shows the results of the CEM testing.

L1 represents the imbalance index of the data, ranging from 0 to 1. The higher L1 is, the more imbalanced the data are. Researchers can evaluate the matching effect based on the change in the L1 value as the more the L1 decreases, the better the matching effect is [61]. Table 3 shows that in the 10-year-old group where students participated in extracurricular arts activities only, 640 samples were kept after CEM, accounting for 55.8% of the total samples, and the L1 dropped from 0.70 before matching to 0.38 after matching. In the 10-year-old group where students participated in extracurricular sports activities only, 870 samples were recorded after CEM, or 67.3% of the total samples, and L1 decreased from 0.62 before matching to 0.40 after matching. In the 10-year-old group where students participated in both extracurricular arts and sports activities, 1406 samples were saved after CEM, reaching 62.2% of the total sample, and L1 declined from 0.71 before matching to 0.41 after matching. In the 15-year-old group where students participated in extracurricular arts activities only, 992 samples were kept after the CEM, amounting to 67.3% of the total sample, and L1 dropped from 0.68 before matching to 0.45 after matching. In the 15-year-old group where students participated in extracurricular sports activities only, 1607 samples were reserved after CEM, accounting for 79.9% of the total sample, and the L1 decreased from 0.60 before matching to 0.43 after matching. In the 15-year-old group where students participated in both extracurricular arts and sports activities, 1535 samples were recorded after CEM, reaching 74.2% of the total samples, and L1 fell from 0.60 before matching to 0.41 after matching. Therefore, the L1 of each experimental group was greatly reduced after CEM, and the matched samples were well preserved.

CEM_weights derived from CEM testing results were brought into the regression, and the weighted regression results are shown in Table 4. After the selection bias was controlled, participating in extracurricular arts activities only had distinctive effects on various dimensions of social and emotional skills of the 10-year-old students. Students who only participated in extracurricular arts activities scored higher in engaging with others (β = 19.10, *p* < 0.01) and open-mindedness (β = 18.68, *p* < 0.01) than those who participated in neither extracurricular arts activities nor sports activities, whereas the two groups of students showed no significant difference in the dimensions of collaboration (β = 3.07, *p* > 0.1), emotional regulation (β = −5.97, *p* > 0.1), and task performance (β = 5.62, *p* > 0.1). Participating in extracurricular sports activities only had different effects on various dimensions of the social and emotional skills of 10-year-old students. Students who only participated in extracurricular sports activities scored significantly higher in three dimensions, collaboration (β = 15.04, *p* < 0.01), engaging with others (β = 18.74, *p* < 0.01), and open-mindedness (β = 17.74, *p* < 0.01), than those who did not participate in extracurricular arts activities or sports activities. But the dimensions of emotional regulation (β = 10.26, *p* > 0.1) and task performance showed (β = 6.10, *p* > 0.1) no significant difference. Students in the 10-year-old group who participated in both extracurricular arts and sports activities performed better in all dimensions of social and emotional skills than those who participated in neither extracurricular arts activities nor sports activities. Specifically, students who participated in both extracurricular arts and sports activities scored significantly higher in all five dimensions of collaboration (β = 28.77, *p* < 0.01), emotional regulation (β = 23.93, *p* < 0.01), engaging with others (β = 41.64, *p* < 0.01), open-mindedness (β = 39.16, *p* < 0.01), and task performance (β = 31.46, *p* < 0.01) than those who participated in neither extracurricular arts activities nor sports activities.

In the 15-year-old group, students who only participated in extracurricular arts activities performed better in all dimensions of social and emotional skills than students who participated in neither extracurricular arts activities nor sports activities. Specifically, students who only participated in extracurricular arts activities achieved better scores in all five dimensions of collaboration (β = 8.90, *p* < 0.1), emotional regulation (β = 12.01, *p* < 0.05), engaging with others (β = 11.96, *p* < 0.01), open-mindedness (β = 15.18, *p* < 0.01), and task performance (β = 9.03, *p* < 0.05) than students who participated in neither extracurricular arts activities nor sports activities. In the 15-year-old group, students who only participated in extracurricular sports activities earned higher scores in all dimensions of social and emotional skills than students who participated in neither extracurricular arts activities nor sports activities. Specifically, students who only participated in extracurricular sports activities scored significantly higher in collaboration (β = 14.82, *p* < 0.01), emotional regulation (β = 15.24, *p* < 0.01), engaging with others (β = 22.21, *p* < 0.01), open-mindedness (β = 10.65, *p* < 0.01), and task performance (β = 8.12, *p* < 0.01) than students who participated in neither extracurricular arts activities nor sports activities. In the 15-year-old group, students who participated in both extracurricular arts and sports activities performed better in all dimensions of social and emotional skills than students who participated in neither extracurricular art nor sports activities. Specifically, students who participated in both extracurricular arts and sports activities scored significantly higher in collaboration (β = 23.67, *p* < 0.01), emotional regulation (β = 23.73, *p* < 0.01), engaging with others (β = 32.55, *p* < 0.01), open-mindedness (β = 27.12, *p* < 0.01), and task performance (β = 20.85, *p* < 0.01) than those who participated in neither extracurricular arts activities nor sports activities.

## 5. Discussion

Participation in extracurricular activities has become an integral part of adolescents’ daily routines. However, prior research has not sufficiently explored the relationships between different types of extracurricular activities and the various dimensions of adolescents’ social and emotional development. This study is one of the initial investigations into the connections between extracurricular activities and the social and emotional competencies of adolescents in our country. This study utilized data from the OECD-SSES in Suzhou, China, to analyze the relationship between participation in extracurricular arts and sports activities and adolescents’ social and emotional skills. The results show that the connection between extracurricular arts and sports activities and the different dimensions of adolescents’ social and emotional skills varies with age. Our findings indicate that participating in different types of extracurricular arts and sports activities at different ages has distinctive effects on various dimensions of adolescents’ social and emotional skills. Specifically, for student groups characterized by similar background factors, including gender, family socioeconomic background, parent–child relationship, and peer relationship, in the 10-year-old group, students who only participate in extracurricular arts activities score significantly higher in the dimensions of engaging with others and open-mindedness than those who participate in neither extracurricular art nor sports activities. Students who only participate in extracurricular sports activities score significantly higher in the three dimensions of collaboration, engaging with others, and open-mindedness than those who do not participate in either extracurricular sports activities or arts activities. Meanwhile, students who participate in both extracurricular sports activities and arts activities achieve significantly higher scores than those who participate in neither extracurricular sports activities nor arts activities. In the 15-year-old group, students who only participate in extracurricular arts activities score significantly higher in all five dimensions of social and emotional skills than those who do not participate in extracurricular arts and sports activities. Students who only participate in extracurricular sports activities score higher in all five dimensions of social and emotional skills than those who do not participate in extracurricular arts and sports activities. Students who participate in both extracurricular sports and arts activities score higher in all five dimensions of social and emotional skills than those who do not participate in extracurricular arts and sports activities. This study confirms that participation in extracurricular arts and sports activities is significantly positively correlated with the enhancement in adolescents’ social and emotional skills, which has been previously established in the literature [22,23]. Additionally, this study reveals that different dimensions of social and emotional skills are variably related to different types of extracurricular arts and sports activities, a topic that has been less explored in prior research.

Since the beginning of this century, scholars from various countries have conducted a lot of research on how to improve the new-era skills such as non-cognitive skills, soft skills, social and emotional skills, and 21st-century skills, and how these new-era skills affect personal development. However, China, for a long time, has lacked relevant large-scale empirical research evidence in this regard. This study, after controlling for sample selection bias, confirms that in the Chinese context, participation in extracurricular arts and sports activities is significantly associated with the overall enhancement in adolescents’ social and emotional skills. This finding aligns with the conclusions of earlier studies conducted in other countries [26,40,63].

To prepare their children to be more competitive among their peers, many Chinese parents begin to engage their children in various extracurricular activities at a very early age [64], including artistic training in music, painting, and dance. In general, participating in extracurricular arts activities can significantly improve skills such as engaging with others and open-mindedness of students in the 10-year-old and 15-year-old groups, which is in accordance with the conclusions drawn by Clift and Hancox [65]. Because of this, participating in extracurricular sports and arts activities allows students to build up and improve their skills of engaging with others and open-mindedness [66,67].

Furthermore, this study revealed a meaningful positive association between involvement in extracurricular sports activities and the improvement in social and emotional skills in both 10-year-old and 15-year-old students. Sports activity has become the most popular extracurricular activity among students [21,42]. A study conducted by Blomfield and Barber points out that adolescents who participate in sports activities score significantly higher than those who do not participate in extracurricular activities in terms of self-perception, self-social perception, and self-academic perception [30]. Similarly, research by Barber, Eccles, and Stone also shows that participation in extracurricular activities has a positive impact on students’ development [68]. In particular, participation in extracurricular sports is associated with higher academic outcomes and better interpersonal relationships. Studies have also pointed out that adolescents participating in sports activities regularly have lower levels of depression, anxiety, loneliness, and suicidal tendencies [68]. Participating in sports can help young students improve themselves, which is usually accompanied by acquiring learning abilities such as autonomy, responsibility, perseverance, and self-control [66,69]. Participating in extracurricular arts and sports activities can by and large help young people build new social networks, and actively engage with more peers and elders. Their participation in an interactive learning environment enhances adolescents’ self-awareness and learning interest and improves their musical rhythm intelligence and bodily–kinesthetic intelligence, leading to better social and emotional skills. Meanwhile, the high social support that adolescents feel and receive from their peers and elders in activities boosts their self-confidence and self-efficacy and enhances their prosocial behaviors [22,24,70]. In addition, participating in extracurricular arts and sports activities is often a process of dealing with problems and confronting challenges. In such a process, adolescents can exercise their ability to resist setbacks and solve problems by using imagination and endurance. At the same time, when cooperating with their peers in extracurricular activities, they understand and embrace interpersonal disparities in a more friendly way that helps to improve their social and emotional skills [63,71]. Therefore, when parents help students arrange extracurricular time, they can allocate more time to extracurricular arts activities and sports activities, instead of limiting the time to extracurricular subject tutoring only.

The attention of researchers is increasingly drawn towards understanding the interplay between different types of extracurricular activities and the development of children and adolescents [50,72,73]. Informed by the definition of synergy by Sirower [74], this study considers that “synergy” happens when the expected benefit a student reaps from participating in two extracurricular activities is greater than the sum of the expected benefits from only participating in extracurricular sports activities and the expected benefits from only participating in extracurricular arts activities. After controlling for relevant impact factors, this study reveals that participating in extracurricular sports activities and arts activities has a synergistic effect on promoting the development of various dimensions of social and emotional skills of adolescents. Regardless of whether the students are in the 10-year-old group or the 15-year-old group, students participating in both extracurricular sports activity and arts activities have higher gains in each dimension of social and emotional skills than those who only participate in extracurricular sports activities or arts activities. Similar conclusions are drawn by Feldman and Matjasko [29] and Blomfield and Barber [30].

Understandably, participating in a variety of extracurricular arts and sports activities at the same time provides adolescents with more opportunities to develop, which in turn helps them cultivate better social and emotional skills [30]. This reflects the importance of promoting students’ participation in extracurricular arts and sports activities in the context of the “double reduction” policy (The “double reduction” policy, introduced by the Chinese government in 2021, is a significant educational reform aimed at reducing the academic burdens on students concerning homework and extracurricular activities. The primary goal is to foster students’ holistic development, reduce the stress associated with an exam-focused education, and ease the financial burden on families regarding their children’s education. Specifically, the “double reduction” policy includes two main components: first, reducing the amount of homework assigned by schools to ensure that students have enough time for relaxation and other activities; second, the strict regulation of extracurricular training institutions, including limits on training duration, content, and fees, thus reducing families’ reliance on external tutoring services. The implementation of the “double reduction” policy has led to substantial changes in the education sector, prompting schools and training institutions to re-evaluate and adapt their educational practices. Simultaneously, this policy encourages parents and society to prioritize the physical and mental well-being and individualized development of students over a singular focus on academic achievement. Through these initiatives, the Chinese government aims to create a more equitable, efficient, and compassionate education system [75].) When implementing the “double reduction” policy to ease the burden of excessive homework and off-campus tutoring for students, we should not simply define the “after-school service” as “after-school learning service”. Instead, we should reserve sufficient time for extracurricular activities, enrich the variety of after-school activities, and encourage students to participate in various types of such activities. Additionally, schools and local governments should follow the relevant education policy and recruit qualified physical education and arts teachers to ensure the regular implementation of curricular and extracurricular sports and arts activities.

In previous studies, different extracurricular activities and different dimensions of social and emotional skills were often aggregated into a single measurement, resulting in a loss of information on the impact of a single type of activity on individual dimensions of social and emotional skills [23,26]. In this study, with the relevant factors controlled, students who participate in different types of extracurricular arts and sports activities are grouped while the five dimensions of social and emotional skills are maintained. Compared to students who do not participate in any extracurricular arts or sports activities, those who participate in either extracurricular arts or sports activities, as well as those who participate in both, show significant differences in various dimensions of social and emotional skills. The latter three groups exhibit significantly better performance in social and emotional skills, particularly in the areas of collaboration, emotional regulation, and task performance, dimensions related to collaboration, emotional regulation, and task proficiency. Extracurricular activities are generally divided into two types: the formal ones and the informal ones [76]. As the name suggests, formal extracurricular activities are mostly carried out regularly, under the supervision of teachers, and usually accompanied by formal examinations and assessments. Informal extracurricular activities, on the other hand, place more emphasis on freedom, improvisation, and innovation, with neither rigorous organization and structure nor accountability and evaluation [77]. In China, 10-year-old students are in primary school, and unlike 15-year-old students, their extracurricular activities are mostly decided and supervised by parents. Parents and teachers are more involved in arranging extracurricular arts activities for 10-year-old students than for 15-year-old students. Because of this, the extracurricular arts activities for the 10-year-old group students tend to be more like formal extracurricular activities. Such involvement and arrangement from parents and teachers can easily weaken the role that extracurricular arts activities play to improve students’ team responsibilities and skills of engaging with others and foster their skills of task performance, collaboration, and emotional regulation. In contrast, 15-year-old students are in high school and have more freedom and choices when participating in extracurricular activities than 10-year-old students. Extracurricular activities for this age group of students can better help build up their social and emotional skills in various dimensions.

The findings of this study extend the application and practice of the Theory of Multiple Intelligences (TMI). Traditionally, TMI has primarily focused on the development of individuals’ abilities in various domains. This study broadens the theory to include the domain of social and emotional skills, demonstrating that sports and arts activities are not only closely related to students’ development in specific intelligence areas but also significantly enhance their social and emotional skills. Furthermore, the results show that students who participate in sports and arts activities exhibit significant advantages in various dimensions of social and emotional skills. This highlights the complementary role of sports and arts activities in promoting the holistic development of students, providing new empirical support for TMI.

This study has some limitations. First, it only examined the broad relationship between arts and sports activities and the improvement in adolescents’ social and emotional skills. Future research could build on this study to explore how specific sports and arts activities relate to the enhancement in these skills. Second, this study initially found significant differences in social and emotional skills across different age groups. Future research could further investigate the reasons for these age-related differences. Third, this study found a significant relationship between family socioeconomic background, peer relationships, parent–child relationships, and the improvement in social and emotional skills. Future research could explore potential causal or moderating effects among these variables. Lastly, since the data used in this study primarily come from surveys of Chinese students, the conclusions may be influenced by the socioeconomic and cultural context of China and may not be directly applicable to other countries. Future research could analyze survey data from different countries to compare the effects of school cooperation and competition on student development across various national and cultural contexts.

In conclusion, this study confirms the positive effects that participation in extracurricular arts and sports activities exerts on adolescents’ social and emotional skills. Moreover, participating in extracurricular arts and sports activities has a synergistic effect on adolescents’ social and emotional skills. Unlike the previous research that combines different dimensions of social and emotional skills into one variable, this study maintains the five dimensions of social and emotional skills for testing and keeps the extracurricular sports activities and arts activities as separate types. Even more intriguingly, we departed from the conventional approach in prior research, which combined various dimensions of social and emotional skills into a single construct. Instead, we maintained the integrity of the five dimensions assessed for social and emotional skills and, upon separating extracurricular sports activities from extracurricular arts activities, discerned differences in the relationships between distinct types of extracurricular pursuits and the various dimensions of social and emotional skills among adolescents.

## Figures and Tables

**Table 1 behavsci-14-00541-t001:** Descriptive Statistics.

Age Group	10-Year-Old Group (N = 3617; Girls = 1650)				15-Year-Old Group (N = 3596; Girls = 1750)			
Activities Group	Arts Activities Only (N = 606; Girls = 380)				Arts Activities Only (N = 495; Girls = 383)			
Statistical Indicators	Min	Max	Mean	SD	Min	Max	Mean	SD
Collaboration	423.50	901.28	646.74	90.56	426.01	889.78	586.46	69.02
Emotional regulation	172.46	920.90	556.18	88.57	217.00	783.41	508.42	74.20
Engaging with others	360.40	871.35	571.80	71.26	368.21	759.64	526.69	56.12
Open-mindedness	370.58	911.84	624.67	85.19	415.93	862.12	584.96	65.87
Task performance	256.22	883.09	618.34	88.47	381.90	848.00	569.26	68.70
Gender	0.00	1.00	0.37	0.48	0.00	1.00	0.23	0.42
Family socioeconomic background	−1.73	2.40	0.33	0.78	−1.48	2.52	0.42	0.76
Peer relationship	10.54	67.27	48.49	14.52	10.54	67.27	46.62	11.04
Parent–child relationship	35.82	93.45	48.36	13.24	35.82	93.45	57.90	10.76
Activities Group	Sports activities only (N = 752; Girls = 166)				Sports activities only (N = 1032; Girls = 302)			
Statistical Indicators	Min	Max	Mean	SD	Min	Max	Mean	SD
Collaboration	403.37	891.42	657.18	91.70	358.40	886.49	596.05	68.40
Emotional regulation	349.14	931.82	576.20	89.32	222.20	927.85	526.90	69.56
Engaging with others	405.07	819.28	582.06	68.28	299.26	799.35	543.55	58.61
Open-mindedness	425.35	901.60	632.54	84.18	420.50	891.35	582.90	69.17
Task performance	398.56	884.30	622.20	87.27	393.13	883.61	576.23	65.52
Gender	0.00	1.00	0.78	0.42	0.00	1.00	0.71	0.46
Family socioeconomic background	−1.91	2.58	0.11	0.82	−1.97	3.05	0.11	0.77
Peer relationship	10.54	67.27	49.06	13.41	10.54	67.27	45.60	10.03
Parent–child relationship	35.82	93.45	48.40	13.37	35.82	93.45	58.05	11.10
Activities Group	Both arts and sports activities (N = 1719; Girls = 924)				Both arts and sports activities (N = 1089; Girls = 581)			
Statistical Indicators	Min	Max	Mean	SD	Min	Max	Mean	SD
Collaboration	382.78	912.78	689.62	100.17	258.74	883.21	611.25	79.93
Emotional regulation	277.02	923.88	602.90	103.98	215.14	921.90	536.69	72.10
Engaging with others	338.64	888.33	608.00	78.15	371.02	835.95	557.33	61.80
Open-mindedness	472.18	910.13	663.01	95.23	385.38	896.47	606.97	77.40
Task performance	289.46	883.61	656.09	97.71	355.64	882.74	590.37	76.20
Gender	0.00	1.00	0.46	0.50	0.00	1.00	0.47	0.50
Family socioeconomic background	−2.05	3.34	0.57	0.84	−2.19	3.34	0.50	0.81
Peer relationship	10.54	67.27	53.24	14.01	10.54	67.27	48.19	11.43
Parent–child relationship	35.82	93.45	46.53	13.41	35.82	93.45	56.55	11.81
Activities Group	Not arts nor sports activities (N = 540; Girls = 180)				Not arts nor sports activities (N = 980; Girls = 484)			
Statistical Indicators	Min	Max	Mean	SD	Min	Max	Mean	SD
Collaboration	388.55	889.78	632.79	99.43	368.06	883.21	573.25	68.17
Emotional regulation	254.34	919.91	549.94	91.90	245.82	875.19	500.67	70.20
Engaging with others	312.48	819.09	552.97	69.81	217.28	813.78	512.08	58.23
Open-mindedness	394.52	893.06	599.70	82.52	353.54	893.06	565.86	65.30
Task performance	408.59	882.05	599.53	89.90	322.43	854.05	559.20	64.34
Gender	0.00	1.00	0.67	0.47	0.00	1.00	0.51	0.50
Family socioeconomic background	−2.44	2.19	−0.15	0.76	−2.15	3.02	0.07	0.83
Peer relationship	10.54	67.27	46.15	14.76	10.54	67.27	44.44	10.83
Parent–child relationship	35.82	93.45	51.28	14.06	35.82	93.45	59.75	11.84

**Table 2 behavsci-14-00541-t002:** Multiple Linear Regression Results.

10-Year-Old Group (Arts Activities Only) (N = 1146)						15-Year-Old Group (Arts Activities Only) (N = 1475)					
Dependent Variable	Collaboration	Emotional Regulation	Engaging with Others	Open-Mindedness	Task Performance	Dependent Variable	Collaboration	Emotional Regulation	Engaging with Others	Open-Mindedness	Task Performance
Independent variable	1.54	−4.84	9.32 **	13.97 ***	6.26	Independent variable	5.82 *	6.41 *	9.58 ***	12.29 ***	6.36 *
	(0.30)	(−0.96)	(2.44)	(2.97)	(1.26)		(1.71)	(1.67)	(3.22)	(3.52)	(1.80)
Gender	−5.94	5.95	3.11	9.89 **	−1.68	Gender	2.80	22.20 ***	4.10	6.26 *	11.16 ***
	(−1.21)	(1.23)	(0.82)	(2.14)	(−0.34)		(0.89)	(6.22)	(1.40)	(1.89)	(3.43)
Family socioeconomic background	−1.07	5.88 *	7.70 ***	13.03 ***	7.00 **	Family socioeconomic background	−1.02	0.35	2.51	8.81 ***	2.77
	(−0.36)	(1.94)	(3.20)	(4.56)	(2.25)		(−0.51)	(0.17)	(1.47)	(4.21)	(1.34)
Peer relationship	3.18 ***	2.07 ***	1.94 ***	1.89 ***	1.86 ***	Peer relationship	2.85 ***	1.74 ***	1.74 ***	1.77 ***	1.46 ***
	(18.84)	(11.26)	(13.93)	(10.86)	(9.45)		(16.74)	(9.34)	(11.83)	(9.84)	(8.44)
Parent–child relationship	−1.28 ***	−1.77 ***	−0.76 ***	−1.12 ***	−1.49 ***	Parent–child relationship	−1.26 ***	−1.97 ***	−0.81 ***	−0.87 ***	−1.45 ***
	(−7.01)	(−8.91)	(−5.25)	(−6.70)	(−7.78)		(−8.70)	(−11.17)	(−6.85)	(−5.73)	(−9.82)
Constant	555.70 ***	541.84 ***	501.46 ***	565.20 ***	592.76 ***	Constant	520.63 ***	530.25 ***	481.12 ***	535.15 ***	574.87 ***
	(40.68)	(35.14)	(45.89)	(41.42)	(37.01)		(43.47)	(39.02)	(49.42)	(42.51)	(47.85)
F	97.46	59.11	61.67	55.94	43.78	F	77.95	49.34	46.19	41.89	39.53
Prob > F	0.00	0.00	0.00	0.00	0.00	Prob > F	0.00	0.00	0.00	0.00	0.00
R2	0.31	0.23	0.23	0.21	0.19	R2	0.28	0.21	0.17	0.16	0.15
10-year-old group (sports activities only) (N = 1292)						15-year-old group (sports activities only) (N = 2012)					
Dependent variable	Collaboration	Emotional regulation	Engaging with others	Open-mindedness	Task performance	Dependent variable	Collaboration	Emotional regulation	Engaging with others	Open-mindedness	Task performance
Independent variable	10.49 **	11.36 **	18.16 ***	18.35 ***	10.26 **	Independent variable	16.54 ***	15.85 ***	25.83 ***	10.84 ***	9.84 ***
	(2.25)	(2.50)	(5.11)	(4.25)	(2.22)		(6.25)	(5.54)	(10.34)	(3.81)	(3.61)
Gender	−4.55	10.17 **	7.74 **	9.30 **	−4.13	Gender	3.70	25.04 ***	10.17 ***	9.15 ***	13.54 ***
	(−0.87)	(2.03)	(2.05)	(1.99)	(−0.83)		(1.38)	(8.66)	(4.15)	(3.19)	(4.93)
Family socioeconomic background	2.12	6.96 **	7.43 ***	14.37 ***	9.12 ***	Family socioeconomic background	0.88	1.20	6.16 ***	10.98 ***	3.84 **
	(0.77)	(2.60)	(3.51)	(5.40)	(3.37)		(0.50)	(0.66)	(4.02)	(5.99)	(2.18)
Peer relationship	3.32 ***	2.35 ***	1.93 ***	2.08 ***	2.17 ***	Peer relationship	2.97 ***	1.78 ***	1.75 ***	1.90 ***	1.67 ***
	(19.57)	(13.72)	(14.59)	(12.77)	(12.39)		(18.10)	(10.63)	(12.00)	(11.15)	(10.40)
Parent–child relationship	−1.43 ***	−1.76 ***	−0.86 ***	−1.25 ***	−1.44 ***	Parent–child relationship	−1.21 ***	−1.91 ***	−0.78 ***	−1.02 ***	−1.40 ***
	(−8.22)	(−9.82)	(−6.66)	(−8.49)	(−8.42)		(−9.67)	(−13.13)	(−6.77)	(−7.31)	(−10.71)
Constant	556.33 ***	525.70 ***	503.90 ***	563.88 ***	577.14 ***	Constant	511.60 ***	522.73 ***	475.12 ***	537.04 ***	561.87 ***
	(42.58)	(37.61)	(50.09)	(48.54)	(40.75)		(46.36)	(46.45)	(47.58)	(46.37)	(49.33)
F	108.72	82.32	81.99	71.58	66.27	F	108.70	84.72	86.11	58.47	62.96
Prob > F	0.00	0.00	0.00	0.00	0.00	Prob > F	0.00	0.00	0.00	0.00	0.00
R2	0.32	0.26	0.25	0.25	0.22	R2	0.29	0.24	0.21	0.18	0.18
10-year-old group (both arts and sports activities) (N = 2259)						15-year-old group (both arts and sports activities) (N = 2069)					
Dependent variable	Collaboration	Emotional regulation	Engaging with others	Open-mindedness	Task performance	Dependent variable	Collaboration	Emotional regulation	Engaging with others	Open-mindedness	Task performance
Independent variable	19.60 ***	19.62 ***	29.01 ***	32.35 ***	22.47 ***	Independent variable	23.07 ***	23.73 ***	34.77 ***	26.48 ***	18.43 ***
	(4.25)	(4.38)	(8.40)	(7.58)	(5.09)		(7.90)	(7.97)	(13.58)	(8.72)	(6.38)
Gender	−10.49 ***	4.15	2.89	5.38	−7.22 **	Gender	8.25 ***	23.10 ***	10.20 ***	9.33 ***	16.21 ***
	(−2.92)	(1.10)	(1.00)	(1.53)	(−1.99)		(2.94)	(8.29)	(4.16)	(3.19)	(5.65)
Family socioeconomic background	7.26 ***	9.19 ***	11.55 ***	14.40 ***	12.38 ***	Family socioeconomic background	−0.87	2.11	4.33 ***	10.38 ***	4.36 **
	(3.24)	(4.00)	(6.60)	(6.65)	(5.55)		(−0.48)	(1.21)	(2.82)	(5.50)	(2.47)
Peer relationship	3.23 ***	2.53 ***	1.91 ***	2.21 ***	2.29 ***	Peer relationship	3.14 ***	1.65 ***	1.85 ***	1.91 ***	1.86 ***
	(23.97)	(17.85)	(17.88)	(16.28)	(16.37)		(20.72)	(11.21)	(14.68)	(12.15)	(12.21)
Parent–child relationship	−1.47 ***	−2.03 ***	−1.01 ***	−1.26 ***	−1.57 ***	Parent–child relationship	−1.21 ***	−1.90 ***	−0.65 ***	−1.05 ***	−1.41 ***
	(−10.69)	(−13.12)	(−9.06)	(−9.88)	(−10.99)		(−9.29)	(−13.16)	(−6.07)	(−7.55)	(−10.76)
Constant	567.07 ***	535.56 ***	516.30 ***	560.60 ***	580.94 ***	Constant	502.108 ***	529.18 ***	463.20 ***	538.33 ***	552.44 ***
	(52.01)	(44.72)	(60.24)	(53.66)	(49.70)		(46.59)	(46.36)	(51.96)	(46.36)	(49.92)
F	222.12	168.73	176.19	158.84	153.73	F	140.95	112.23	132.36	96.82	83.64
Prob > F	0.00	0.00	0.00	0.00	0.00	Prob > F	0.00	0.00	0.00	0.00	0.00
R2	0.33	0.28	0.27	0.26	0.26	R2	0.33	0.27	0.27	0.23	0.22

Note: The value in parenthesis is the T value; ***, **, and * represent the significance levels of 1%, 5%, and 10%, respectively.

**Table 3 behavsci-14-00541-t003:** CEM Testing.

		10-Year-Old Group			15-Year-Old Group		
		Control Group	Experimental Group	Total	Control Group	Experimental Group	Total
Arts activities only	Total Sample	540	606	1146	981	495	1476
	Matching Sample	311	329	640	594	398	992
	Unmatched Sample	229	277	506	387	97	484
	Before Matching L1	0.70			0.68		
	After Matching L1	0.38			0.45		
Sports activities only	Total Sample	540	752	1292	980	1032	2012
	Matching Sample	363	507	870	762	845	1607
	Unmatched Sample	177	245	422	218	187	405
	Before Matching L1	0.62			0.60		
	After Matching L1	0.40			0.43		
Both arts and sports activities	Total Sample	540	1719	2259	980	1089	2069
	Matching Sample	402	1004	1406	726	809	1535
	Unmatched Sample	138	715	853	254	280	534
	Before Matching L1	0.71			0.60		
	After Matching L1	0.41			0.41		

**Table 4 behavsci-14-00541-t004:** Weighted Regression Results of CEM.

10-Year-Old Group (Arts Activities Only) (N = 640)						15-Year-Old Group (Arts Activities Only) (N = 992)					
Dependent Variable	Collaboration	Emotional Regulation	Engaging with Others	Open-Mindedness	Task Performance	Dependent Variable	Collaboration	Emotional Regulation	Engaging with Others	Open-Mindedness	Task Performance
Independent variable	3.07	−5.97	19.10 ***	18.68 ***	5.62	Independent variable	8.90 *	12.01 **	11.96 ***	15.18 ***	9.033 **
	(0.40)	(−0.72)	(3.52)	(2.46)	(0.79)		(1.94)	(2.57)	(3.24)	(3.62)	(1.93)
Control variable	YES	YES	YES	YES	YES	Control variable	YES	YES	YES	YES	YES
Constant	525.53 ***	554.98 ***	503.28.03 ***	517.17 ***	560.76 ***	Constant	531.16 ***	498.58 ***	483.43 ***	536.67 ***	568.08 ***
	(23.39)	(25.01)	(26.14)	(21.63)	(23.99)		(25.65)	(24.19)	(30.49)	(29.91)	(26.47)
F	36.49	17.60	25.67	25.98	26.41	F	32.14	17.22	23.44	19.73	19.73
Prob > F	0.00	0.00	0.00	0.00	0.00	Prob > F	0.00	0.00	0.00	0.00	0.00
R^2^	0.31	0.21	0.21	0.25	0.23	R^2^	0.29	0.15	0.17	0.15	0.13
10-year-old group (sports activities only) (N = 870)						15-year-old group (sports activities only) (N = 1607)					
Dependent variable	Collaboration	Emotional regulation	Engaging with others	Open-mindedness	Task performance	Dependent variable	Collaboration	Emotional regulation	Engaging with others	Open-mindedness	Task performance
Independent variable	15.04 **	10.26	18.74 ***	17.74 ***	6.10	Independent variable	14.82 ***	15.24 ***	22.21 ***	10.65 ***	8.12 ***
	(2.45)	(1.42)	(4.04)	(2.66)	(0.91)		(4.15)	(3.50)	(6.73)	(2.97)	(2.30)
Control variable	YES	YES	YES	YES	YES	Control variable	YES	YES	YES	YES	YES
Constant	524.88 ***	501.14 ***	489.67 ***	534.04 ***	547.66 ***	Constant	496.19 ***	501.13 ***	478.58 ***	526.05 ***	541.50 ***
	(27.53)	(29.15)	(31.15)	(29.42)	(26.85)		(29.83)	(34.97)	(35.30)	(36.15)	(36.03)
F	67.02	41.85	46.92	38.82	42.90	F	51.83	38.95	42.69	34.58	33.28
Prob > F	0.00	0.00	0.00	0.00	0.00	Prob > F	0.00	0.00	0.00	0.00	0.00
R^2^	0.32	0.26	0.25	0.28	0.26	R^2^	0.27	0.18	0.17	0.17	0.16
10-year-old group (both arts and sports activities) (N = 1406)						15-year-old group (both arts and sports activities) (N = 1535)					
Dependent variable	Collaboration	Emotional regulation	Engaging with others	Open-mindedness	Task performance	Dependent variable	Collaboration	Emotional regulation	Engaging with others	Open-mindedness	Task performance
Independent variable	28.77 ***	23.93 ***	41.64 ***	39.16 ***	31.46 ***	Independent variable	23.67 ***	23.73 ***	32.55 ***	27.12 ***	20.85 ***
	(3.21)	(2.13)	(8.37)	(4.17)	(3.65)		(5.36)	(6.26)	(10.03)	(6.34)	(4.89)
Control variable	YES	YES	YES	YES	YES	Control variable	YES	YES	YES	YES	YES
Constant	542.00 ***	536.61 ***	497.72 ***	542.11 ***	568.76 ***	Constant	491.00 ***	501.54 ***	456.31 ***	511.04 ***	536.10 ***
	(30.76)	(31.17)	(37.52)	(31.81)	(34.14)		(33.47)	(32.91)	(37.49)	(31.07)	(29.68)
F	114.33	82.99	89.87	86.41	91.33	F	64.84	53.79	71.55	49.34	36.99
Prob > F	0.00	0.00	0.00	0.00	0.00	Prob > F	0.00	0.00	0.00	0.00	0.00
R^2^	0.30	0.24	0.24	0.25	0.25	R^2^	0.29	0.21	0.23	0.19	0.19

Note: The value in parenthesis is the T value; ***, **, and * represent the significance levels of 1%, 5%, and 10%, respectively.

## Data Availability

The data that support the findings of this study are available in OECD at Data from the Survey on Social and Emotional Skills. These data were derived from the following resources available in the public domain: https://www.oecd.org/education/ceri/social-emotional-skills-study/data.htm (accessed on 16 September 2021).

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
