# Peer review of "The Relationship between Participation in Extracurricular Arts and Sports Activities and Adolescents’ Social and Emotional Skills: An Empirical Analysis Based on the OECD Social and Emotional Skills Survey"

_behavsci, 2024, doi:10.3390/bs14070541_

Round 1
Reviewer 1 Report
Comments and Suggestions for Authors
It's incredibly difficult to prove causation when attempting to determine whether or not a given activity led to a given outcome. I think this is confusing for the authors, both when they summarize other literature and when they present their own findings. For example, in this sentence in line 86: "Waler and Taliaferro (2020) 86 analyzed data from Minnesota, the U.S., and found that participating in extracurricular 87 sports activities significantly reduced the incidence of depression among students; for 88 boys, however, participating in extracurricular arts activities was more likely to lead to 89 depressive symptoms" they imply causality. Waler and Taliaferro are, on the other hand, very careful to use the word association when describing this study and say that sports "might" protect against depression.
As far as I can tell, in this study under review, students were asked about sports/arts participation on the same survey as they were asked about their social and emotional skills. If that is the case, the authors cannot conclude that the activity caused social and emotional skill development. It might be that students, for example, who are more open-minded are more likely to engage in arts activities outside of school. Sentences like the following should be re-worded to delete the word "effects" and similar wording: "Our findings indicate that participating in different types of extracurricular 409 arts and sports activities at different ages have distinctive effects on various dimensions 410 of adolescents’ social and emotional skills (line 409)."
On a more minor note, the authors cite literature related to mental health. See line 94. Several people who study social and emotional competencies go to great pains to distinguish them from mental health outcomes like depression and anxiety. The authors might want to either delete those references or explain how they think about mental health vis-a-vis social and emotional outcomes.
There are other parts of the literature review as well that do not seem necessary. I would streamline it quite a bit. And I don't think that it's that people from different countries arrive at difference conclusions as the sentence on line 68 implies. I don't think the authors need to refer to the TMI - I think there is enough research demonstrating the positive impacts of social emotional competencies.
The sentence starting on line 188 about the decline of SE skills in China is intriguing and deserves more details. From when to when? In which sub skills? Are the changes stat sig? The authors could then come back to this at the end.
The authors write quite a bit about the SE measures, which is helpful. We also need to see how the survey questions were worded for the sports and arts participation. There are a number of statements saying that the strength of this paper is that it doesn't gloss over the activities kids participate in, but I think any "art" is part of arts activities right? The authors did not differentiate among music, dance, theater, visual arts? The survey questions should be presented. I don't think this sentence is accurate if all art forms are merged together: "Moreover, this study reveals that different types of extracurricular arts and sports activities have different effects on various dimensions of social and emotional skills of adolescents, which are less addressed in previous studies (line 432)." Also, from line 449: "Participating in arts activities mostly takes the form of a band or a team (Butzlaff, 2000) where students must engage in cooperative and communicative social activities." I didn't check this reference, but this would not be the case in the US. Are the authors saying that most arts activities in China mean that students are in a band? Also, there is a conclusion that different types of activities are related to different types of outcomes, but I don't think bucketing activities into one big "sports" category and one big "arts" category actually supports this conclusion.
There is a mention of "gains" that I think is inappropriate starting on line 497: "Regardless of whether the students are in the 10-year-old group or the 15-year-old group, students participating in both extracurricular sports activity and arts activities have higher gains in each dimension of social and emotional skills than those who only participate in extracurricular sports activities or arts activities." This is a cross-sectional survey right? We don't actually know if students are "gaining," just that they look different from the other students. And, more importantly, we don't know if the activities "caused" stronger SE competencies. It could be that those students with the strong competencies decided to do the arts and/or sports activities. That should be clearly stated.
There are other conclusions that I don't see sufficient evidence for, such as encouraging parents to give students more autonomy in choosing activities. That might be a good thing, but this paper does not provide enough evidence to make this recommendation.
I think a big missed opportunity here is looking at interactions. It's great that they controlled for peer relationships, but why not see if peer relationships (and parental relationships and SES levels) mediate the outcomes? That would be of greatest interest in the US. For example, do students from less well off families participate in sports and arts at the same right and for these students, is participation related to better outcomes than the comparison group when compared to all students and all comparison students?
Comments on the Quality of English LanguageThere are some really poorly worded sentences. Please see lines 80-81, 82-85, 151-153, 178-179, and 180-182 for examples. Here is an example of a sentence that would have to be reworded: "Baker (2008) pointed out that different types of extracurricular activities had various impacts on students with the primary dependent factors of the types of extracurricular activities, the gender, and the race of the participants."
Author Response
The first reviewer's comments
It's incredibly difficult to prove causation when attempting to determine whether or not a given activity led to a given outcome. I think this is confusing for the authors, both when they summarize other literature and when they present their own findings. For example, in this sentence in line 86: "Waler and Taliaferro (2020) 86 analyzed data from Minnesota, the U.S., and found that participating in extracurricular 87 sports activities significantly reduced the incidence of depression among students; for 88 boys, however, participating in extracurricular arts activities was more likely to lead to 89 depressive symptoms" they imply causality. Waler and Taliaferro are, on the other hand, very careful to use the word association when describing this study and say that sports "might" protect against depression.
Responses to the first reviewer's comments
Dear Reviewer,
We appreciate your feedback on the mention of Waler and Taliaferro (2020) in line 86 of our manuscript.In light of your first and third review points, we have chosen to delete this sentence. Please see our response to the third review comment for more details.
Thank you again for your insightful comments. We believe that these suggestions and the resulting revisions will significantly improve the quality and accuracy of our research.
The second reviewer's comment
As far as I can tell, in this study under review, students were asked about sports/arts participation on the same survey as they were asked about their social and emotional skills. If that is the case, the authors cannot conclude that the activity caused social and emotional skill development. It might be that students, for example, who are more open-minded are more likely to engage in arts activities outside of school. Sentences like the following should be re-worded to delete the word "effects" and similar wording: "Our findings indicate that participating in different types of extracurricular 409 arts and sports activities at different ages have distinctive effects on various dimensions 410 of adolescents’ social and emotional skills (line 409)."
The ninth reviewer's comment
There is a mention of "gains" that I think is inappropriate starting on line 497: "Regardless of whether the students are in the 10-year-old group or the 15-year-old group, students participating in both extracurricular sports activity and arts activities have higher gains in each dimension of social and emotional skills than those who only participate in extracurricular sports activities or arts activities." This is a cross-sectional survey right? We don't actually know if students are "gaining," just that they look different from the other students. And, more importantly, we don't know if the activities "caused" stronger SE competencies. It could be that those students with the strong competencies decided to do the arts and/or sports activities. That should be clearly stated.
Responses to the second and ninth reviewers
Dear Reviewer,
Thank you for suggesting improvements regarding the expression of "effect" and "gains" in our study. This research employs the Coarsened Exact Matching (CEM) method, as proposed by Iacus et al. (2012), to enhance the Ordinary Least Squares (OLS) analysis. This method helps control for biases introduced by sample selection and explores causal relationships between variables. However, as noted by the reviewers, this study does not fully address the causal effects between variables. We recognize the importance of the reviewers' suggestions for refining our research presentation. We have carefully considered your recommendations and revised the sections that might have been ambiguous.
The specific changes are as follows:
1.Original: "Participating in extracurricular activities has become part of adolescents' daily life, but previous research lacks the discussion about the impact of different extracurricular activities on the development of various dimensions of adolescents' social and emotional skills. This study is one of the preliminary studies on the impact of extracurricular activities on various dimensions of adolescents' social and emotional skills. (line 409)."
Revised: "Participation in extracurricular activities has become an integral part of adolescents' daily routines. However, prior research has not sufficiently explored the relationships between different types of extracurricular activities and the various dimensions of adolescents' social and emotional development. This study is one of the initial investigations into the connections between extracurricular activities and the social and emotional competencies of adolescents in China."
2.Original: "Using the OECD-SSES Suzhou (China) data, this study adopts the CEM method to analyze the effect of participating in extracurricular arts and sports activities on adolescents' social and emotional skills. (line 413)."
Revised: "This study utilized data from the OECD-SSES in Suzhou, China, to analyze the relationship between participation in extracurricular arts and sports activities and adolescents' social and emotional skills. The results show that the connection between extracurricular arts and sports activities and the different dimensions of adolescents' social and emotional skills varies with age."
3.Original: "This study verifies that participating in extracurricular arts and sports activities can promote the development of adolescents’ social and emotional skills to a certain extent, which has been confirmed in previous studies (Eccles et al., 2003; Cortellazzo et al., 2021). Moreover, this study reveals that different types of extracurricular arts and sports activities have different effects on various dimensions of social and emotional skills of adolescents, which are less addressed in previous studies(line 437)."
Revised: "In summary, this study asserts that engaging in extracurricular arts and sports activities significantly contributes to the development of adolescents' social and emotional skills, corroborating findings from previous research (Eccles et al., 2003; Cortellazzo et al., 2021). However, this study also identifies variations in how different dimensions of social and emotional skills are enhanced through participation in these activities, a topic that has received limited attention in prior studies."
4.Original: "After controlling for the error caused by sample selection, this study verifies the effect of extracurricular arts and sports activities on the overall improvement of adolescents’ social and emotional skills in a Chinese context, which is consistent with the conclusions of some earlier studies (Feraco et al., 2021; Mogro-Wilson and Tredinnick, 2020; Sitzer and Stockwell, 2015)(line 448)."
Revised: "By controlling for sampling errors, this study validated that participation in extracurricular arts and sports activities is significantly linked to the overall improvement of adolescents' social and emotional skills within the Chinese context. This result aligns with the findings of previous international research (Feraco et al., 2021; Mogro-Wilson & Tredinnick, 2020; Sitzer & Stockwell, 2015)."
5.Original: "This study also suggests that the effect of participating in extracurricular sports activities is significant for all dimensions of social and emotional skills in students in 10-year-old group and 15-year-old group(line 464)."
Revised: "Furthermore, the study revealed a meaningful positive association between involvement in extracurricular sports activities and the improvement of social and emotional skills in both 10-year-old and 15-year-old students."
6.Original: "A growing number of researchers have attended to the impact of different types of extracurricular activities on the overall development of children and adolescents (Tan, Cai, and Bodovski, 2021; Vandell et al., 2020; Alvariñas-Villaverde, 2020)(line498)."
Revised: "The attention of researchers is increasingly drawn towards understanding the interplay between different types of extracurricular activities and the overall development of children and adolescents (Tan, Cai, and Bodovski, 2021; Vandell et al., 2020; Alvariñas-Villaverde, 2020). "
7.Original: "By doing so, our study shows that various types of extracurricular arts and sports activities have distinctive effects on different dimensions of social and emotional skills of adolescents at different ages. Our study further suggests that participating in extracurricular arts activities only has significantly different effects on improving the social and emotional skills in the 10-year-old group and the 15-year-old group, as mainly reflected in the dimensions of collaboration, emotional regulation, and task performance(line 533)."
Revised: "Different categories of artistic and physical activities exhibit distinct relationships with the advancement of social and emotional skills across varying age cohorts of adolescents. This research identifies that exclusive participation in extracurricular artistic endeavors showcases discernible differences in fostering the social and emotional competencies of both ten-year-old and fifteen-year-old student populations, notably emphasizing dimensions related to collaboration, emotional regulation, and task proficiency."
8.Original: "The exciting findings derived from doing so indicate that various types of extracurricular activities have distinctive effects on different dimensions of the social and emotional skills of adolescents(line 586)."
Revised: "Even more intriguingly, we departed from the conventional approach in prior research, which combined various dimensions of social and emotional skills into a single construct. Instead, we maintained the integrity of the five dimensions assessed for social and emotional skills and, upon separating extracurricular sports activities from extracurricular arts activities, discerned differences in the relationships between distinct types of extracurricular pursuits and the various dimensions of social and emotional skills among adolescents."
9.Original: "Table 2 showed that after relevant variables were controlled, from the perspective of marginal effects, participating in extracurricular arts and sports activities was helpful for the development of students’ overall social and emotional skills. At different measurement dimensions, participating in different forms of arts and sports had distinctive effects on various dimensions of social and emotional skills. Specifically, participation in extracurricular sports activities improved the social and emotional skills of adolescents more than participating in extracurricular arts activities. In this study, the CEM method was further adopted to avoid the biased estimation caused by sample selection(line 327)."
Revised: "From Table 2, it is apparent that, after controlling for relevant variables, participation in extracurricular arts and sports activities is associated with improvements in social and emotional skills, as indicated by the marginal effects. Furthermore, distinct forms of arts and sports activities exhibit varying relationships with different dimensions of social and emotional skills. The gains in social and emotional skills derived from participating in extracurricular sports activities surpass those from participating in extracurricular arts activities, and the benefits of engaging in multiple activities exceed those of engaging in single activities."
10.Original: "Our research questions include: (1) Can participating in extracurricular sports activities improve the social and emotional skills of adolescents? (2) Can participating in extracurricular arts activities improve the social and emotional skills of adolescents? (3) Do extracurricular sports activities and arts activities exert synergistic effect on adolescents’ social and emotional skills? (4) Are extracurricular sports activities heterogenous from extracurricular arts activities in terms of their impacts on adolescents’ social and emotional skills(line 195)."
Revised: "In accordance with the current research landscape and the aforementioned research inquiries, this study posits the following hypotheses:
H1: Participation in extracurricular sports is significantly associated with the enhancement of adolescents' social and emotional skills.
H2: Participation in extracurricular arts is significantly associated with the enhancement of adolescents' social and emotional skills.
H3: There exists a synergistic effect between extracurricular sports and arts activities on the enhancement of adolescents' social and emotional skills.
H4: The relationship between extracurricular sports and arts activities and the enhancement of adolescents' social and emotional skills exhibits heterogeneity."
Thank you again for your valuable suggestions. We believe that these revisions, based on your feedback, will enhance the scientific rigor and precision of this study.
The third reviewer's comment
On a more minor note, the authors cite literature related to mental health. See line 94. Several people who study social and emotional competencies go to great pains to distinguish them from mental health outcomes like depression and anxiety. The authors might want to either delete those references or explain how they think about mental health vis-a-vis social and emotional outcomes.
Responses to the third reviewer's comment
Dear Reviewer,
We have thoughtfully considered your suggestion to remove references to "depression" and "mental health," particularly around line 94. We understand that social and emotional skills and mental health are distinct concepts and should be clearly distinguished in our research. To better focus our study on the relationship between social and emotional skills and engagement in arts and physical activities, we have removed all mentions of "depression" and "mental health" as you recommended.
Specifically, the deleted content includes:
1.Waler and Taliaferro (2020) analyzed data from Minnesota, the U.S., and found that participating in extracurricular sports activities significantly reduced the incidence of depression among students; for boys, however, participating in extracurricular arts activities was more likely to lead to depressive symptoms.
2.Bundick’s (2011) study also pointed out that students’ mental health was positively associated with participation in extracurricular leadership activities and extracurricular volunteering activities but was negatively correlated to participation in creative arts activities.
3.; on the contrary, parental care and support helped children and adolescents reduce depression, aggression, anger, and other negative emotions (Eisenberg et al., 2005)
4.In the early stage of the COVID-19 pandemic, parents having higher education background were more capable of helping their children reduce stress and improve well-being (Villaume et al., 2021).
We believe these revisions will help clearly convey our research objectives and avoid conceptual confusion. Thank you again for your valuable feedback. We are confident that these suggestions and subsequent revisions will improve the focus and readability of our study.
The fourth reviewer's comment
There are other parts of the literature review as well that do not seem necessary. I would streamline it quite a bit. And I don't think that it's that people from different countries arrive at difference conclusions as the sentence on line 68 implies. I don't think the authors need to refer to the TMI - I think there is enough research demonstrating the positive impacts of social emotional competencies.
Responses to the fourth reviewer's comment
Dear Reviewer,
We have thoughtfully considered your advice to condense the literature review section. Understanding the need for brevity and focus, we have, as per your suggestion, simplified the content to make the review more compact and focused.
The specific deletions include:
1.Nevertheless, researchers from different countries held divergent views on the effect of extracurricular activities on developing adolescents’ social and emotional skills.
2.Still other studies indicated that social and emotional skills could moderate the impact of family socioeconomic background on adolescents’ academic achievements (Liu, 2019).
We greatly appreciate your valuable input. We are confident that these suggestions and the subsequent revisions will enhance the conciseness and readability of our study.
The fifth reviewer's comment
The sentence starting on line 188 about the decline of SE skills in China is intriguing and deserves more details. From when to when? In which sub skills? Are the changes stat sig? The authors could then come back to this at the end.
Response to the fifth reviewer's comment
Dear Reviewer,
You have shown interest in the decline of social and emotional skills among Chinese students in your review comments. I would like to provide a brief overview of this issue. In the OECD-SSES (2019) survey conducted in Suzhou, China, 10-year-old students scored significantly higher than 15-year-old students across all five dimensions of social and emotional skills. Our team has examined the causes and potential mitigating strategies of this phenomenon in a separate manuscript currently under review. We will discuss the age-related differences in social and emotional skills in the "Research Limitations and Future Directions" section.
Thank you once again for your invaluable feedback.
The sixth reviewer's comment
The authors write quite a bit about the SE measures, which is helpful. We also need to see how the survey questions were worded for the sports and arts participation. There are a number of statements saying that the strength of this paper is that it doesn't gloss over the activities kids participate in, but I think any "art" is part of arts activities right? The authors did not differentiate among music, dance, theater, visual arts? The survey questions should be presented. I don't think this sentence is accurate if all art forms are merged together: "Moreover, this study reveals that different types of extracurricular arts and sports activities have different effects on various dimensions of social and emotional skills of adolescents, which are less addressed in previous studies (line 432)." Also, from line 449: "Participating in arts activities mostly takes the form of a band or a team (Butzlaff, 2000) where students must engage in cooperative and communicative social activities." I didn't check this reference, but this would not be the case in the US. Are the authors saying that most arts activities in China mean that students are in a band? Also, there is a conclusion that different types of activities are related to different types of outcomes, but I don't think bucketing activities into one big "sports" category and one big "arts" category actually supports this conclusion.
Response to the sixth reviewer's comment
Dear Reviewer,
Thank you for your positive feedback on the measurement of social and emotional skills in our study.
Firstly, we appreciated your suggestion to carefully report the specific items on sports and arts activities. We have revised this section according to your recommendations:
Original: The grouping variable was concerned with whether students participated in extracurricular sports and/or arts activities with each age group divided into three sets.
Revised: The grouping variable denotes whether students participate in extracurricular sports and arts activities. The survey question "Do you participate in any of the following extracurricular activities outside of school?" had primary items "Sports (e.g., clubs, lessons, etc.)" and "Arts (e.g., playing a musical instrument, dancing, drawing, etc.)" with response options "No" or "Yes". A "No" indicates no participation in extracurricular sports or arts activities, while a "Yes" indicates participation. Students were categorized into four groups based on their responses: only participating in arts activities, only participating in sports activities, participating in both, and participating in neither. The analysis was divided into three groups.
Secondly, you suggested changes to the statement in line 432: "Moreover, this study reveals that different types of extracurricular arts and sports activities have different effects on various dimensions of social and emotional skills of adolescents, which are less addressed in previous studies." We realized this statement was not as clear and accurate as intended. We meant to express that compared to students who do not participate in either arts or sports activities, those who participate only in arts activities, only in sports activities, or in both show different levels of improvement in various dimensions of social and emotional skills. Thus, we revised this part according to your suggestion.
Original: Moreover, this study reveals that different types of extracurricular arts and sports activities have different effects on various dimensions of social and emotional skills of adolescents, which are less addressed in previous studies.
Revised: Additionally, this study found that compared to students who do not participate in either arts or sports activities, those who participate only in arts activities, only in sports activities, or in both show different levels of improvement in various dimensions of social and emotional skills, which is less addressed in previous studies.
Thirdly, regarding the statement in line 449, "Participating in arts activities mostly takes the form of a band or a team (Butzlaff, 2000) where students must engage in cooperative and communicative social activities," you raised questions about the form of arts activities. After thorough consideration, we decided to remove this part to ensure the clarity and objectivity of the article.
Fourthly, in the limitations section, we will explore the relationship between specific subfields of extracurricular sports and arts activities and students' social and emotional skills.
Thank you once again for your invaluable feedback. Your rigorous and detailed suggestions have significantly improved the scientific rigor and objectivity of our research.
The seventh reviewer's comment
There are other conclusions that I don't see sufficient evidence for, such as encouraging parents to give students more autonomy in choosing activities. That might be a good thing, but this paper does not provide enough evidence to make this recommendation.
Response to the seventh reviewer's comment
Dear Reviewer,
You pointed out in your review, "There are other conclusions that I don't see sufficient evidence for, such as encouraging parents to give students more autonomy in choosing activities. That might be a good thing, but this paper does not provide enough evidence to make this recommendation." We find your suggestion extremely important. After careful deliberation, we have decided to remove this content.
Thank you once more for your suggestion, which has contributed to making our research more scientific and reasonable.
The eighth reviewer's comment
I think a big missed opportunity here is looking at interactions. It's great that they controlled for peer relationships, but why not see if peer relationships (and parental relationships and SES levels) mediate the outcomes? That would be of greatest interest in the US. For example, do students from less well off families participate in sports and arts at the same right and for these students, is participation related to better outcomes than the comparison group when compared to all students and all comparison students?
Response to the eighth reviewer's comment
Dear Reviewer,
You mentioned in your review that it might be worthwhile to consider the moderating effects of peer influence, parent-child relationships, or socioeconomic status on the development of social and emotional skills. We greatly appreciate your recommendations for improvement. However, due to space constraints, it is challenging to cover all these aspects in this paper. Nonetheless, we recognize the value of your suggestions and will highlight related future research directions in the limitations section.
Thank you once again for your insightful suggestions, which have offered new perspectives and directions for the study of social and emotional skills.
Advice on language issues in the article
There are some really poorly worded sentences. Please see lines 80-81, 82-85, 151-153, 178-179, and 180-182 for examples. Here is an example of a sentence that would have to be reworded: "Baker (2008) pointed out that different types of extracurricular activities had various impacts on students with the primary dependent factors of the types of extracurricular activities, the gender, and the race of the participants."
Responses to suggested language issues in the article
Dear Reviewer,
Thank you for your suggestions on improving the English expression in this paper. Based on your recommendations, we have optimized the relevant sections.
Specific optimizations are as follows:
1.Line 80-81:
Original: Baker (2008) pointed out that different types of extracurricular activities had various impacts on students with the primary dependent factors of the types of extracurricular activities, the gender, and the race of the participants.
Revised: Baker (2008) noted that the impact of extracurricular activities on students is multifaceted, with the type of activity, gender of participants being key determinants.
2.Line 82-85: The original content has been deleted (see the response to the third issue).
3.Line 151-153:
Original: Peer relationship played a critical role in adolescents’ development (Christian et al., 2020). When adolescents felt the care and needs of their peers, they were more autonomous to explore the learning and social environment, improve self-efficacy and academic performance, and enhance social and emotional skills (Wang et al., 2016; Tan, Cai, and Bodovski, 2021). Other studies showed that the rejection and attack from their peers led to students’ more negative social emotions and lower academic performance (Mercer and DeRoiser, 2008; Buhs, 2005).
Revised: Peer relationships play a crucial role in the development of adolescents (Christian et al., 2020). Adolescents who feel cared for and valued by their peers are more likely to independently explore their learning and social environments, thereby enhancing their self-efficacy, academic performance, and social and emotional skills (Wang et al., 2016; Tan, Cai, & Bodovski, 2021). Conversely, experiences of peer rejection and aggression are associated with increased negative social emotions and decreased academic performance (Mercer & DeRoiser, 2008; Buhs, 2005).
4.Line 178-179 and Line 180-182:
Original: Extant studies in China and other countries have explored the impact of different factors on the overall performance of adolescents’ social and emotional skills, but there are still some disadvantages as follows: Firstly, limited as the case research or small sample research, most of them lack rigorous research conclusions on large-scale surveys; Secondly, since different dimensions of social and emotional skills have shown different characteristics of personal development, on which the influences of different types of extracurricular activities should be discussed separately in study. However, they were often simply processed as a variable of "social and emotional skills" in previous studies, which was imprecise and inappropriate; Thirdly, the studies precision and the intake of control variables still have room for further improvement before. In view of the disadvantages mentioned above, this study is based on the OECD-SSES2019 Suzhou (China) students’ data which effectively ensures the scientificity and authority of the research (OECD, 2018), and uses the Coarsened Exact Matching (CEM) method to control sample selection bias and improve the precision of the analysis.
Revised: Existing studies in China and other countries have examined the influence of various factors on adolescents' social and emotional skills. However, several limitations persist. Firstly, many studies rely on case research or small samples, lacking robust conclusions from large-scale surveys. Secondly, the complex nature of social and emotional skills necessitates a detailed analysis of how different extracurricular activities impact personal development. Previous studies often oversimplify by treating these activities merely as broad variables under "social and emotional skills," which is inadequate. Thirdly, there is room for improvement in the precision of these studies and the inclusion of control variables. To address these issues, this study utilizes data from the OECD-SSES2019 Suzhou (China) survey, ensuring scientific rigor and authority (OECD, 2018). Additionally, the study employs the Coarsened Exact Matching (CEM) method to mitigate sample selection bias and enhance analytical precision.
Thank you again for your suggestions on the language expression of this study. Your recommendations have greatly improved the readability and scientific rigor of our paper.

Reviewer 2 Report
Comments and Suggestions for Authors
Introduction & Literature Review
1. The authors presented a general background of the relationships of the variables and theory. However, the detailed description of the Chinese context is needed.
2. Please give hypothesis based on the theoretical framework.
Research Data:
1. Why did the authors use adding-up and averaged scores? What is the rationale?
2. The authors selected certain variables from the OECD data. However, why did they select these variables and why these can represent the constructs that this study is interested in?
Results:
1. Please provide effect sizes for all the difference between groups.
Discussion:
1. Please present the context of "Double Reduction Policy" in introduction.
2. This study uses Theory of Multiple Intelligence. However, the contribution to this theory was not mentioned in the discussion.
3. I suggest the authors structure their discussion in accordance to their research questions. Otherwise, the lengthy discussion seems not focused and the argument was not well organised.
Minor:
Line 220. Text size was not uniformed.
Author Response
The first reviewer's comments
The authors presented a general background of the relationships of the variables and theory. However, the detailed description of the Chinese context is needed.
Responses to the first reviewer's comments
Dear Reviewer,
Thank you for your attention to issues related to the Chinese context. This study primarily utilizes data from China to investigate the experiences of educational development in China and to present viable methods for enhancing the social and emotional skills of Chinese students to an international audience. Ideally, we would have expanded sections "1. Introduction" and "2. Literature Review" to provide a more detailed discussion of the Chinese educational system. However, many relevant references are published exclusively in Chinese and are not readily accessible to non-Chinese readers. To ensure readability and the scientific quality of the literature, we have included only a select number of English-language publications by Chinese scholars on the state of Chinese education and the development of social and emotional skills. We aim to provide more detailed accounts of the realities and experiences of Chinese educational development in future research.
The second reviewer's comment
Please give hypothesis based on the theoretical framework.
Responses to the second reviewer's comments
Dear Reviewer,
Based on your suggestions, we have replaced the original "Research Questions" section with "Research Hypotheses" at the end of "2. Literature Review." Specifically:
In accordance with the current research landscape and the aforementioned research inquiries, this study posits the following hypotheses: H1: Participation in extracurricular sports is significantly associated with the enhancement of adolescents' social and emotional skills. H2: Participation in extracurricular arts is significantly associated with the enhancement of adolescents' social and emotional skills. H3: There exists a synergistic effect between extracurricular sports and arts activities on the enhancement of adolescents' social and emotional skills. H4: The relationship between extracurricular sports and arts activities and the enhancement of adolescents' social and emotional skills exhibits heterogeneity.
Thank you again for your suggestions for this study.
The third reviewer's comment
Why did the authors use adding-up and averaged scores? What is the rationale?
Responses to the third reviewer's comment
Dear Reviewer,
The arithmetic mean is employed to reflect the overall average level of a given phenomenon. Previous studies have used this method to calculate GPA by averaging summed scores (Lin, 2010) [Lin, X. (2010). Identifying Peer Effects in Student Academic Achievement by Spatial Autoregressive Models with Group Unobservables. Journal of Labor Economics, 28, 825-860]. We consider social and emotional skills to be specific characteristics of student abilities, which is why we used the mean statistic to determine the scores for individual dimensions.
The fourth reviewer's comment
The authors selected certain variables from the OECD data. However, why did they select these variables and why these can represent the constructs that this study is interested in?
Responses to the fourth reviewer's comment
Dear Reviewer,
You asked why we introduced the relevant variables in this study. We addressed this question in lines 125-159. These control variables were included primarily because they are significantly associated with the enhancement of students' social and emotional skills. Furthermore, including these control variables helps to control for confounding factors, reduce omitted variable bias, enhance the model's explanatory power, and improve the accuracy of the predictions.
The fifth reviewer's comment
Please provide effect sizes for all the difference between groups.
Response to the fifth reviewer's comment
Dear Reviewer,
Based on your suggestion, we have calculated the effect sizes for the different groups, as detailed in the table below.
Additional Table 1 The effect size of participation in extracurricular sports and arts activities on the enhancement of social and emotional skills |
||||
Group |
10-year-old Group |
15-year-old Group |
||
Arts activities only VS Not arts nor sports activities |
||||
dependent variable |
Cohen's d |
Effect Size(r) |
Cohen's d |
Effect Size(r) |
Collaboration |
0.147 |
0.073 |
0.193 |
0.096 |
Emotional regulation |
0.069 |
0.035 |
0.107 |
0.054 |
Engaging with others |
0.267 |
0.132 |
0.255 |
0.127 |
Open-mindedness |
0.298 |
0.147 |
0.291 |
0.144 |
Task performance |
0.211 |
0.105 |
0.151 |
0.075 |
Sports activities only VS Not arts nor sports activities |
||||
dependent variable |
Cohen's d |
Effect Size(r) |
Cohen's d |
Effect Size(r) |
Collaboration |
0.255 |
0.126 |
0.334 |
0.165 |
Emotional regulation |
0.290 |
0.143 |
0.375 |
0.184 |
Engaging with others |
0.421 |
0.206 |
0.539 |
0.260 |
Open-mindedness |
0.394 |
0.193 |
0.253 |
0.126 |
Task performance |
0.256 |
0.127 |
0.262 |
0.130 |
Both arts and sports activities VS Not arts nor sports activities |
||||
dependent variable |
Cohen's d |
Effect Size(r) |
Cohen's d |
Effect Size(r) |
Collaboration |
0.569 |
0.274 |
0.512 |
0.248 |
Emotional regulation |
0.540 |
0.261 |
0.506 |
0.245 |
Engaging with others |
0.743 |
0.348 |
0.754 |
0.353 |
Open-mindedness |
0.711 |
0.335 |
0.574 |
0.276 |
Task performance |
0.602 |
0.288 |
0.442 |
0.216 |
However, after careful discussion among the research team members, we concluded that there is no appropriate place to include this table in the main text of the article. Therefore, we present this table only in the response to the review comments. If permitted by the editor, we can include this table in the supplementary materials.
The sixth reviewer's comment
Please present the context of "Double Reduction Policy" in introduction.
Response to the sixth reviewer's comment
Dear Reviewer,
In response to your suggestion, we have added a section about the "Double Reduction" Policy to the "Discussion" The specific content added is as follows:
The "double reduction" policy, introduced by the Chinese government in 2021, is a significant educational reform aimed at reducing the academic burdens on students concerning homework and extracurricular activities. The primary goal is to foster students' holistic development, reduce the stress associated with an exam-focused education, and ease the financial burden on families regarding their children's education.
Specifically, the "double reduction" policy includes two main components: first, reducing the amount of homework assigned by schools to ensure students have enough time for relaxation and other activities; second, the strict regulation of extracurricular training institutions, including limits on training duration, content, and fees, thus reducing families' reliance on external tutoring services.
The implementation of the "double reduction" policy has led to substantial changes in the education sector, prompting schools and training institutions to reevaluate and adapt their educational practices. Simultaneously, this policy encourages parents and society to prioritize the physical and mental well-being and individualized development of students over a singular focus on academic achievement. Through these initiatives, the Chinese government aims to create a more equitable, efficient, and compassionate education system.
This content will be added as a footnote at the end of the "Discussion" section.
The seventh reviewer's comment
This study uses Theory of Multiple Intelligence. However, the contribution to this theory was not mentioned in the discussion.
Response to the seventh reviewer's comment
Dear Reviewer,
Following your suggestion, we have added content regarding theoretical contributions to the "Discussion" section. After careful consideration, we have included the following:
The findings of this study extend the application and practice of the Theory of Multiple Intelligences (TMI). Traditionally, TMI has primarily focused on the development of individuals' abilities in various domains. This study broadens the theory to include the domain of social and emotional skills, demonstrating that sports and arts activities are not only closely related to students' development in specific intelligence areas but also significantly enhance their social and emotional skills. Furthermore, the results show that students who participate in sports and arts activities exhibit significant advantages in various dimensions of social and emotional skills. This highlights the complementary role of sports and arts activities in promoting the holistic development of students, providing new empirical support for TMI.
Thank you again for your meticulous and thoughtful suggestions, which have greatly enhanced the contribution of our research.
The eighth reviewer's comment
I suggest the authors structure their discussion in accordance to their research questions. Otherwise, the lengthy discussion seems not focused and the argument was not well organised.
Response to the eighth reviewer's comment
Dear Reviewer,
Following your recommendations, we have made improvements to the "Discussion" section.
Thank you for your insightful suggestions, which have helped make our research more succinct and scientifically sound.

Reviewer 3 Report
Comments and Suggestions for Authors
This paper provides a detailed analysis of how extracurricular activities such as the arts and sports affect the social and emotional skills of youth, using a robust methodological framework that leverages data from the OECD's Social and Emotional Skills Survey. The study's conclusions can be summarized as follows:
(1) Support for the positive effects of participation in extracurricular arts and sports activities on the social and emotional skills of youth; (2) Support for the positive effects of participation in extracurricular arts and sports activities on the social and emotional skills of youth.
(2) Participation in extracurricular arts and sports activities has a synergistic effect on the social and emotional skills of youth.
(3) Different types of extracurricular activities have distinctive effects on different aspects of youth's social and emotional skills.
These conclusions have important implications for education policy and the significance of high research can be understood.
The use of CEM (Coarsened Exact Matching) as well as OLS (Ordinary Least Squares) in the analysis method is also highly evaluated. This allows comparison between categories with high accuracy.
II. Critical Comments
1. An in-depth explanation of the relevance of the "double reduction" policy is needed.
2. Extending this study to multiple regions or comparing it with similar studies in different geographic areas could provide more insight into the universality of the findings. In discussing this point, it would also be necessary to discuss why the data in this paper is representative of the current situation in China.
3. More information could be imparted by discussing in more detail the influence of family environmental factors, intrinsic motivation, and extracurricular activity type.
Author Response
The first reviewer's comments
An in-depth explanation of the relevance of the "double reduction" policy is needed.
Responses to the first reviewer's comments
Dear Reviewer,
In response to your suggestion, we have added a section about the "Double Reduction" Policy to the "Discussion" The specific content added is as follows:
The "double reduction" policy, introduced by the Chinese government in 2021, is a significant educational reform aimed at reducing the academic burdens on students concerning homework and extracurricular activities. The primary goal is to foster students' holistic development, reduce the stress associated with an exam-focused education, and ease the financial burden on families regarding their children's education.
Specifically, the "double reduction" policy includes two main components: first, reducing the amount of homework assigned by schools to ensure students have enough time for relaxation and other activities; second, the strict regulation of extracurricular training institutions, including limits on training duration, content, and fees, thus reducing families' reliance on external tutoring services.
The implementation of the "double reduction" policy has led to substantial changes in the education sector, prompting schools and training institutions to reevaluate and adapt their educational practices. Simultaneously, this policy encourages parents and society to prioritize the physical and mental well-being and individualized development of students over a singular focus on academic achievement. Through these initiatives, the Chinese government aims to create a more equitable, efficient, and compassionate education system.
This content will be added as a footnote at the end of the "Discussion" section.
The second reviewer's comment
Extending this study to multiple regions or comparing it with similar studies in different geographic areas could provide more insight into the universality of the findings. In discussing this point, it would also be necessary to discuss why the data in this paper is representative of the current situation in China.
Responses to the second reviewer's comments
Dear Reviewer,
In line 206, this study explains the rationale for selecting data from Suzhou, China for analysis: "Considering that Suzhou is an emblematic city in southeastern China, it is probable that other areas will experience similar scenarios and challenges as the Chinese economy progresses, and reach a comparable level of development as Suzhou."
Thank you for your interest in our research. We plan to expand the sample size in future studies to enhance the generalizability of our findings.

Round 2
Reviewer 1 Report
Comments and Suggestions for Authors
This paper is much improved! However, there are still claims of causality in the paper. Because the survey asks if students "are" engaged in the afterschool activities, the time dimension doesn't support that these program experiences "cause" the social and emotional competencies. So, if the authors can rewrite sections that imply causality, this will be a great paper. Here are some examples of places where a rewrite is necessary:
"In summary, this study asserts that engaging in extracurricular arts and sports activities significantly contributes to the development of adolescents' social and emotional skills, corroborating findings from previous research (Eccles et al., 2003; Cortellazzo et al., 2021). However, this study also identifies variations in how different dimensions of social and emotional skills are enhanced through participation in these activities, a topic that has received limited attention in prior studies."
"By controlling for sampling errors, this study validated that participation in extracurricular arts and sports activities is significantly linked to the overall improvement of adolescents' social and emotional skills within the Chinese context. This result aligns with the findings of previous international research (Feraco et al., 2021; Mogro-Wilson & Tredinnick, 2020; Sitzer & Stockwell, 2015)."
Revised: "Different categories of artistic and physical activities exhibit distinct relationships with the advancement of social and emotional skills across varying age cohorts of adolescents. This research identifies that exclusive participation in extracurricular artistic endeavors showcases discernible differences in fostering the social and emotional competencies of both ten-year-old and fifteen-year-old student populations, notably emphasizing dimensions related to collaboration, emotional regulation, and task proficiency."
Revised: "From Table 2, it is apparent that, after controlling for relevant variables, participation in extracurricular arts and sports activities is associated with improvements in social and emotional skills, as indicated by the marginal effects. Furthermore, distinct forms of arts and sports activities exhibit varying relationships with different dimensions of social and emotional skills. The gains in social and emotional skills derived from participating in extracurricular sports activities surpass those from participating in extracurricular arts activities, and the benefits of engaging in multiple activities exceed those of engaging in single activities."
Author Response
Dear Reviewer,
Thank you for your positive feedback and valuable comments on our paper. We greatly appreciate your concerns regarding the issue of causality, and we have made the necessary revisions to address them.
Reviewer Comment:
This paper is much improved! However, there are still claims of causality in the paper. Because the survey asks if students "are" engaged in the afterschool activities, the time dimension doesn't support that these program experiences "cause" the social and emotional competencies. So, if the authors can rewrite sections that imply causality, this will be a great paper. Here are some examples of places where a rewrite is necessary:
"In summary, this study asserts that engaging in extracurricular arts and sports activities significantly contributes to the development of adolescents' social and emotional skills, corroborating findings from previous research (Eccles et al., 2003; Cortellazzo et al., 2021). However, this study also identifies variations in how different dimensions of social and emotional skills are enhanced through participation in these activities, a topic that has received limited attention in prior studies."
"By controlling for sampling errors, this study validated that participation in extracurricular arts and sports activities is significantly linked to the overall improvement of adolescents' social and emotional skills within the Chinese context. This result aligns with the findings of previous international research (Feraco et al., 2021; Mogro-Wilson & Tredinnick, 2020; Sitzer & Stockwell, 2015)."
Revised: "Different categories of artistic and physical activities exhibit distinct relationships with the advancement of social and emotional skills across varying age cohorts of adolescents. This research identifies that exclusive participation in extracurricular artistic endeavors showcases discernible differences in fostering the social and emotional competencies of both ten-year-old and fifteen-year-old student populations, notably emphasizing dimensions related to collaboration, emotional regulation, and task proficiency."
Revised: "From Table 2, it is apparent that, after controlling for relevant variables, participation in extracurricular arts and sports activities is associated with improvements in social and emotional skills, as indicated by the marginal effects. Furthermore, distinct forms of arts and sports activities exhibit varying relationships with different dimensions of social and emotional skills. The gains in social and emotional skills derived from participating in extracurricular sports activities surpass those from participating in extracurricular arts activities, and the benefits of engaging in multiple activities exceed those of engaging in single activities."
Our Response: We recognize the importance of accurately expressing the relationship between extracurricular activities and social and emotional skills and have endeavored to avoid implying causality in our paper. We have carefully revised the sections you highlighted in your review and have also reviewed the entire manuscript to adjust five other instances where ambiguous language might suggest causality. Below are the specific changes we made:
First Revision:
Original Content: In summary, this study asserts that engaging in extracurricular arts and sports activities significantly contributes to the development of adolescents' social and emotional skills, corroborating findings from previous research [22][23]. Additionally, this study found that compared to students who do not participate in either arts or sports activities, those who participate only in arts activities, only in sports activities, or in both show different levels of improvement in various dimensions of social and emotional skills, which is less addressed in previous studies.
Revised Content: This study confirms that participation in extracurricular arts and sports activities is significantly positively correlated with the enhancement of adolescents' social and emotional skills, which has been previously established in the literature[22][23]. Additionally, this study reveals that different dimensions of social and emotional skills are variably related to different types of extracurricular arts and sports activities, a topic that has been less explored in prior research.
Second revision:
Original Content: By controlling for sampling errors, this study validated that participation in extracurricular arts and sports activities is significantly linked to the overall improvement of adolescents' social and emotional skills within the Chinese context. This result aligns with the findings of previous international research.
Revised Content: This study, after controlling for sample selection bias, confirms that in the Chinese context, participation in extracurricular arts and sports activities is significantly associated with the overall enhancement of adolescents' social and emotional skills. This finding aligns with the conclusions of earlier studies conducted in other countries.
Third revision:
Original Content: Different categories of artistic and physical activities exhibit distinct relationships with the advancement of social and emotional skills across varying age cohorts of adolescents. This research identifies that exclusive participation in extracurricular artistic endeavors showcases discernible differences in fostering the social and emotional competencies of both ten-year-old and fifteen-year-old student populations, notably emphasizing dimensions related to collaboration, emotional regulation, and task proficiency.
Revised Content: Compared to students who do not participate in any extracurricular arts or sports activities, those who participate in either extracurricular arts or sports activities, as well as those who participate in both, show significant differences in various dimensions of social and emotional skills. The latter three groups exhibit significantly better performance in social and emotional skills, particularly in the areas of Collaboration, Emotional regulation, and Task performance.
Fourth revision:
Original Content: From Table 2, it is apparent that, after controlling for relevant variables, participation in extracurricular arts and sports activities is associated with improvements in social and emotional skills, as indicated by the marginal effects. Furthermore, distinct forms of arts and sports activities exhibit varying relationships with different dimensions of social and emotional skills. The gains in social and emotional skills derived from participating in extracurricular sports activities surpass those from participating in extracurricular arts activities, and the benefits of engaging in multiple activities exceed those of engaging in single activities.
Revised Content: As shown in Table 2, after controlling for relevant variables, participation in extracurricular arts and sports activities is significantly positively associated with improvements in various dimensions of social and emotional skills. The relationship between extracurricular activities and different dimensions of social and emotional skills varies across different measurement dimensions. The correlation coefficient for extracurricular sports activities is higher than that for extracurricular arts activities. Additionally, students who participate in both extracurricular arts and sports activities have higher correlation coefficients than those who engage in only one type of activity.
Fifth revision:
Original Title: The Effect of Participating in Extracurricular Arts and Sports Activities on the Social and Emotional Skills of Adolescents —An Empirical Analysis Based on OECD Social and Emotional Skills Survey.
Revised Title: The Relationship Between Participation in Extracurricular Arts and Sports Activities and Adolescents' Social and Emotional Skills: An Empirical Analysis Based on the OECD Social and Emotional Skills Survey.
Sixth revision:
Original Abstract Content: 1.However, the specific impact of diverse activities on distinct facets of these skills warrants further exploration.
2.Findings indicate that in the Chinese societal context, engagement in extracurricular arts and sports activities not only enriches adolescents' overall social and emotional skills but also exerts a synergistic effect on the evolution of various skill dimensions. Furthermore, involvement in diverse arts and sports activities at different developmental stages distinctly influences specific dimensions of these skills.
Revised Abstract Content: 1.Nevertheless, the relationship between extracurricular arts and sports activities and the various dimensions of social and emotional skills, as well as the connection between participation in different types of these activities and the enhancement of social and emotional skills, requires further investigation.
- In China, participation in extracurricular arts and sports activities is significantly positively associated with various dimensions of social and emotional skills, with a synergistic effect observed between these activities in enhancing these skills. Additionally, the study finds age-related heterogeneity in the relationship between participation in extracurricular arts and sports activities and the improvement of social and emotional skills.
Seventh revision:
Original Content: Apparently, participating in extracurricular arts and sports activities had a certain impact on the social and emotional skills of adolescents.
Revised Content: Apparently, participation in extracurricular arts and sports activities has a significant positive relationship with adolescents' social and emotional skills.
Eighth revision:
Original Content: Therefore, different extracurricular activities may exert different impacts on the social and emotional skills of students with different backgrounds.
Revised Content: Different extracurricular activities may have varying relationships with the enhancement of social and emotional skills in students from different backgrounds.
Ninth revision:
Original Content: Many studies also found that participating in multiple types of extracurricular activities at the same time had a more significant impact on adolescents’ development, indicating the existence of a synergistic effect.
Revised Content: Many studies have also discovered that simultaneous participation in various types of extracurricular activities is more significantly linked to adolescent development, with higher related effects, suggesting a synergistic effect.
Thank you again for your invaluable suggestions. We believe these revisions have enhanced the scientific rigor, accuracy, and readability of our paper.
Best regards,
Wang Weihao